# Structural Damage Detection under Ambient Excitation Using Symbolic Three-Order Square Matrix Formed by Specific-Interval-Sampled Time-Domain Signals

**DOI:** 10.3390/s24185941

**Published:** 2024-09-13

**Authors:** Shuang Meng, Dongsheng Li

**Affiliations:** 1Department of Civil Engineering, Dalian University of Technology, Dalian 116024, China; mengs@mail.dlut.edu.cn; 2MOE Key Laboratory of Intelligent Manufacturing Technology, Shantou University, Shantou 515063, China; 3Shantou Key Laboratory of Offshore Wind Energy, Guangdong Engineering Center for Structure Safety and Health Monitoring, Department of Civil Engineering, Shantou University, Shantou 515063, China

**Keywords:** structural damage detection, symbolic three-order square matrix, symbolic time-series analysis, ambient excitation

## Abstract

In the structural health monitoring of vibration systems, varying excitation always affects the accuracy of damage identification. The proposed symbolic three-order square matrix damage detection method with the matrix norm as a damage indicator can solve the difficult problem of damage identification under ambient excitation. The new sampling pattern extracts data from signals in the time domain at specific intervals based on the structural properties with the help of the autocorrelation coefficient. Then, the data extracted are converted into symbols and arranged into a three-order square matrix, and the Frobenius norm of the matrix is used for structural damage identification as a reliable damage indicator. In this process, the transmissibility function is employed to eliminate the effects of varying excitation. First, the method was verified by a cracked simply supported beam—a simulated Abaqus model. Then, a wooden truss bridge in the laboratory and an actual engineering scenario under ambient excitation together demonstrated the effectiveness and accuracy of the damage identification method and proved the proposed method to be robust to different types of damage under ambient excitation. Compared with other related methods, this method is more intuitive and efficient.

## 1. Introduction

In structural health monitoring (SHM), damage identification has always been of great significance. With the development of damage identification, most nondestructive damage identification methods can be categorized either as local or global techniques [1,2]. Local damage identification approaches, including visual, acoustic, ultrasonic, and X-ray methods, require that the location of the damage be known in advance and the examined component be accessible for testing. However, these requirements cannot always be guaranteed in most large-scale civil engineering structures. Consequently, global damage identification methods are developed to overcome these shortcomings in the determination of structural damages.

Global structural damage identification methods can also be divided into two categories: static- and dynamic-based damage identification methods. Static-based methods require fewer test facilities and data and are relatively simple, rapid, intuitive, and economical. Xiao et al. [3,4] introduced a stiffness separation method for damage identification in large-scale space truss structures based on static response and optimal static strain sensor placement and achieved effective results. In contrast, dynamic-based global damage identification methods are more flexible to evaluate the state of large complicated structures and have been extensively applied in civil engineering in recent decades. The fundamental theory for vibration-based damage identification [5,6] is that damage-induced changes in physical properties (mass, damping, and stiffness) lead to a detectable variation in modal properties (natural frequencies, mode shapes, mode shape curvature, modal flexibility, modal strain energy, etc.). In other words, damage can be deduced based on the variation in such modal properties. In addition to these basic damage indicators, other relevant parameters that can be used as damage indicators are also derived, including the transmissibility function.

The transmissibility function (TF) was proposed to represent the relationship between different outputs of a system [7] whose mathematical representation can express the characteristics of the system itself, which will not change along with the variation in the input. In other words, it is not sensitive to the excitation of the structure in damage identification. But for now, the TF, as defined, cancels the excitation effects only for a single-point excitation. Zhang et al. [8] used transmittance functions to detect damage on a composite beam, and the optimal frequency range to detect damage was experimentally determined. Zhou et al. [9] proposed a new transmissibility-based damage detection and quantification approach to detect nonlinearity damage. Similar to the transfer function, the TF has been studied in both the time and the frequency domains. In the frequency domain, the amplitude–frequency characteristic curve of the damage-sensitive frequency band is used as a damage indicator to explore the damage state of the structure. However, it is the key and difficult point to find the influential frequency band susceptible to damage, which makes the TF attract more attention in the time domain, where the TF is transformed into the virtual frequency response function (VFRF) by the inverse Fourier transform. As a freely decaying real signal, the VFRF still contains all the damaged information on the entire frequency band, which effectively avoids the band selection problem [10]. This real signal usually has a relatively high dimension, so appropriate signal processing methods are needed for dimensionality reduction. Jiang et al. [11] employed wavelet analysis in signal processing for damage recognition, Li et al. [12] performed structural damage identification under ambient excitation based on wavelet packet analysis, and Ortiz et al. [13] processed time-domain signals for identification with Bayesian optimization; in this context, the symbolic time-series method is also an effective technique.

Symbolic time-series analysis (STSA) is a statistical-based method of signal processing and feature extraction which has been used to detect damage in various fields [14,15,16,17]. Combined with Shannon’s entropy, this method can capture the main features of vibration systems and reduce the dimension of the information collected from vibration signals. The primary steps of the STSA method for damage identification are as follows: The partitioning of the symbol space is carried out first based on the maximum entropy approach, whereby the raw time-series measurements are uniquely mapped into symbol sequences. Then, the symbol sequences are formed by defining a finite-length template to group consecutive symbols, and this template is shifted along the symbol series. Subsequently, each word is encoded into its equivalent code to form a code series. Finally, the probability damage detection vectors are developed according to the relative frequencies of each symbol sequence to account for anomalies [18,19,20]. STSA still has limitations in damage identification because the capture of effective damage information is not accurate enough, the workload is still large, and errors increase with structural complexity. A new sampling pattern based on the structural properties is then proposed to solve the problem, which means that some specific intervals are used for sampling in STSA with the help of some other tools.

In this study, a new damage identification method based on STSA is proposed combined with assistant tools such as the TF, Shannon’s entropy, and the autocorrelation coefficient. In contrast to the disadvantages of the conventional symbolic time-series method, this method can more accurately capture the damaged state of a structure. Section 2 introduces the transformation process of the data from a raw time series to a symbolic three-order square matrix with an explanation of relevant theories. In Section 3, theoretical derivation is carried out to verify the proposed method. In Section 4 and Section 5, a simulation model and an experiment in the laboratory are used to prove the effectiveness and reliability of the proposed method. And in Section 6, the method is further verified by an actual engineering scenario. For all the structures, the types of damage and the excitation mode considered in the identification of structural damage under ambient excitation are different. The conclusions and discussion are drawn in Section 7.

## 2. Signal Processing and Conversion

Typically, sensors located throughout the structure pick up signals generated during vibration. For large experimental structures or practical engineering health monitoring, signal acquisition and application form the premise of damage identification. Generally, the collected signals are speed, acceleration, displacement, force, and so on. Temperature, wind speed, load, and other environmental factors are also monitored. The proposed method for structural damage detection under ambient excitation mainly uses acceleration signals, and the processing and conversion steps of the signals are detailed in this section. The existing theories involved are explained along with the process.

The new sampling pattern to form the three-order matrix is illustrated in Figure 1. To summarize, the TF is calculated from the raw time series, and the VFRF series is obtained by the inverse Fourier transform. The VFRF series is then divided into three partitions based on maximum entropy. We symbolize the VFRF series data according to the partition in which they are located, and the symbols from smallest to largest are 1, 2, 3. After that, we find the special interval by using the autocorrelation coefficient and combine the two symbols separated by the interval into symbol pairs. We drop each pair of symbols into the nine grids, and the number of pairs will represent the entries of the matrix. Figure 1 also explains the operation in detail.

### 2.1. Raw Time Series to VFRF Series

The original signal is usually a time-varying data series collected at a certain sampling frequency called a raw time series. The process of acceleration data acquisition is inevitably affected by environmental factors, which leads to great uncertainty in damage identification.

To solve this problem, the TF is introduced. After the inverse Fourier transform, the VFRF series transformed from the TF is still a time-varying data series, which can eliminate the impact of environmental excitation and reflect the characteristics of the structure.

#### Transmissibility Function

To understand VFRF, the TF should be introduced first. The TF is the ratio of the Fourier transform of responses at two different points on the structure as defined in [21] and can be expressed as
(1)$$TF_{ij}\left(\omega \right)=\frac{X_{i}\left(\omega \right)}{X_{j}\left(\omega \right)},$$
where $X_{i}\left(\omega \right)$ is the fast Fourier transform of structural vibration response $x_{i}\left(t\right)$ at measurement point *i*. $X_{j}\left(\omega \right)$ is the fast Fourier transform of structural vibration response $x_{j}\left(t\right)$ at the reference point *j*. The following relationship can be obtained: (2)$$TF_{ij}\left(\omega \right)=\frac{X_{i}\left(\omega \right)}{X_{j}\left(\omega \right)}=\frac{\sum_{m=1}^{N_{F}}H_{i,m}\left(\omega \right)F_{m}\left(\omega \right)}{\sum_{m=1}^{N_{F}}H_{j,m}\left(\omega \right)F_{m}\left(\omega \right)},$$
where $H_{i,m}\left(\omega \right)$ is the frequency response function with input at point *i* and output at point *m*. $F_{m}\left(\omega \right)$ is the fast Fourier transform of the excitation on the structure at point *m*, and $N_{F}$ is the number of excitation points. If it is single-point excitation acting on point *m*, Equation (Equation 2) can be simplified as
(3)$$TF_{ij}\left(\omega \right)=\frac{X_{i}\left(\omega \right)}{X_{j}\left(\omega \right)}=\frac{H_{i,m}\left(\omega \right)F_{m}\left(\omega \right)}{H_{j,m}\left(\omega \right)F_{m}\left(\omega \right)}=\frac{H_{i,m}\left(\omega \right)}{H_{j,m}\left(\omega \right)}.$$

We use Power Spectrum Density Transmissibility to avoid the influence of loading condition [22]; Equation (Equation 2) can be changed into the following expression:(4)$$\begin{array}{cc} TF_{ij}\left(\omega \right)\; & =\frac{G_{i,j}\left(\omega \right)}{G_{j,j}\left(\omega \right)}=\frac{X_{i}\left(\omega \right)\cdot X_{j}^{*}\left(\omega \right)}{X_{j}\left(\omega \right)\cdot X_{j}^{*}\left(\omega \right)}\\ \; & =\frac{\left(H_{i,m}\left(\omega \right)F_{m}\left(\omega \right)\right)\cdot \left(H_{j,m}^{*}\left(\omega \right)F_{m}^{*}\left(\omega \right)\right)}{\left(H_{j,m}\left(\omega \right)F_{m}\left(\omega \right)\right)\cdot \left(H_{j,m}^{*}\left(\omega \right)F_{m}^{*}\left(\omega \right)\right)}\\ \; & =\frac{H_{i,m}\left(\omega \right)\cdot H_{j,m}^{*}\left(\omega \right)}{H_{j,m}\left(\omega \right)\cdot H_{j,m}^{*}\left(\omega \right)}, \end{array}$$
where $X_{j}^{*}\left(\omega \right)$ is the complex conjugate of $X_{j}\left(\omega \right)$. $G_{i,j}\left(\omega \right)$ is the cross-power spectral density function between response $x_{j}\left(t\right)$ at reference point *j* and response $x_{i}\left(t\right)$ at measurement point *i*. $G_{j,j}\left(\omega \right)$ is the auto-power spectral density function of $x_{j}\left(t\right)$.

The VFRF is the inverse Fourier transform of the TF, which can be expressed as
(5)$$VFRF_{i,j}\left(t\right)=\frac{1}{2\pi }\int_{-\infty }^{+\infty }e^{-i\omega t}TF_{i,j}\left(\omega \right)d\omega .$$

It is worth noting that the position of the measurement point is best chosen away from the reference point to guarantee the effectiveness of the method.

### 2.2. VFRF Series to Symbol Series

The theory of the partitioning and symbolization of a time series comes from symbolic time-series analysis (STSA), which is a relatively mature statistical analysis method. STSA often allows for the capturing of the main features of the original time-series data at a large scale, which can significantly reduce the dimension of the underlying data and their sensitivity to noise contamination. Because of this premise, the statistical features of symbol sequences can be used to describe the dynamic state of the system. The core step of STSA is to discretize the underlying time-series data into a corresponding sequence of symbols based on maximum entropy, which is illustrated below with the help of Figure 1.

#### 2.2.1. Maximum Entropy

The concept of entropy was first introduced by C. Shannon in 1948 [23,24] to characterize the statistical characteristics of symbol series; it is a measure used to describe the complexity and uncertainty of dynamical systems. Generally, the lower the value is, the more orderly the system is, and vice versa. The expression of Shannon’s entropy is approximated as follows: (6)$$H_{m}\left(X\right)=H_{m}\left(p_{1},p_{2},\ldots ,p_{m}\right)=-\sum_{i=1}^{m}p_{i}\log p_{i},$$
where $p_{i}$ denotes the probability of symbol occurrence in the *i*th partition. i∈[1,m]. *m* is the number of partitions.

As observed in Equation (Equation 6), the entropy keeps going down to a minimum as one of the probabilities gets infinitely close to 1; conversely, the maximum entropy occurs when the probabilities of each symbol are equal to each other. We bring Equation (Equation 6) into the current condition, the partition number (*m*) is 3, and $p_{i}$ is the probability that the VFRF series data fall in the *i*th partition as symbol *i*. The maximum entropy of the divided data into three partitions means that the probabilities of all partition symbols should be as equal as possible. Preferably, the number of data is the same in the three partitions.

#### 2.2.2. Partitioning and Symbolization

The partitioning of the symbol space is carried out based on maximum entropy, and symbolization is then performed by assigning symbols to each datum based on the region into which it falls. Thus, the VFRF series are uniquely mapped into a symbol series. Partitioning and symbolization work together to transform the time series from state to symbol space, capture the changing features of the time series, and shield the impact of invalid information. For example, as shown in Figure 1, the VFRF series is divided into three partitions representing the symbols 1, 2, 3 from the bottom to the top. According to the maximum entropy theory, the number of data in each partition is preferably the same. In that case, the VFRF series data are transformed into symbol series S={123211332112332123}, no matter how large the range of variation in the raw series is.

### 2.3. Symbol Series to Symbol Pair Series

The key part of the new sampling pattern to form the three-order square matrix for structural damage identification, as well as the biggest difference from conventional methods, is the choice of interval τ. On the foundation of the original sampling frequency, a suitable interval is selected through the expression of the autocorrelation coefficient, so that the two symbols with a specific interval are combined as a symbol pair.

The process of selecting the interval according to the autocorrelation coefficient is as follows: the autocorrelation coefficient is plotted, and the stable extremum points are found; the corresponding τ is the interval containing rich information on structural characteristics. Generally, there is more than one interval eligible from one operation for certainty. In Figure 1, the interval is already obtained as 8, which means that the first and ninth symbols from the symbol series are combined as 12, and so are the second and tenth ones, third and eleventh ones… Finally, the symbol series becomes symbol pair series $S_{p}=\left\{12\;21\;31\;22\;13\;13\;32\;31\;22\;13\right\}$.

#### Autocorrelation Coefficient

The correlation coefficient [25] in the usual sense is a measure of the correlation between two variables. The coefficient for two variables either positively or negatively correlated is widely used in the study of various fields, as evidenced by various research works. The linear relationship between two continuous variables is mostly expressed as a Pearson product–moment correlation [26], which is typically used for jointly normally distributed data. For non-normally distributed continuous data, ordinal data, or data with relevant outliers, a Spearman rank correlation can be used as a measure of a monotonic association. Other simplified or variant correlation coefficients have also been proposed and applied in many fields. To study the degree of autocorrelation of two values in the same signal at different intervals, the autocorrelation coefficient [27] is used based on the Pearson correlation coefficient to find the suitable interval τ, so that the original signal series are formed into subsequences based on the interval, leading to further research on the properties of the time series. The expression of the autocorrelation coefficient is as follows: (7)$$C\left(\tau \right)=\frac{1}{N-\tau }\sum_{k=1}^{N-\tau }s_{k}s_{k+\tau },$$
where $s_{k}$ is the *k*th discrete datum in the time series, τ is the interval between two chosen data, and *N* is the length of the time series.

### 2.4. Symbol Pair Series to Matrix

The symbol pair series has nine types of data in Figure 1. We put the data into a 3×3 grid wholly according to the current code, as represented in the figure, where 12 is in the first row, second column; then, 21 is in the second row, first column. After all the data have been put in, the numbers of the symbol pairs in each position are shown in the 3×3 blue grid. The three-order square matrix is then formed by matching these numbers to the same positions as the nine grids.

Then, the 3×3 matrix A(τ) is constructed as
$$A=\left[\begin{array}{ccc} a_{11} & a_{12} & a_{13}\\ a_{21} & a_{22} & a_{23}\\ a_{31} & a_{32} & a_{33} \end{array}\right],$$
where $a_{ij}$ is the number of the symbol pairs.

#### Damage Indicator

The three-order square matrix with the new sampling pattern contains rich information on structural vibration characteristics for damage identification, and the Frobenius norm (FN) of the matrix as damage indicator can accurately evaluate the occurrence and degree of structural damage, which is denoted by
(8)$$C\left(\tau \right)=\frac{1}{N-\tau }\sum_{k=1}^{N-\tau }s_{k}s_{k+\tau },$$
(9)$$FN={∥A∥}_{F}=\sqrt{\sum_{i=1}^{m}\sum_{j=1}^{n}{\left|a_{ij}\right|}^{2}},$$
where $a_{ij}$ are the entries of the three-order square matrix, with i,j=1,2,3.

## 3. Theoretical Derivation

### 3.1. Finding Special Intervals

We assume an *F* degree-of-freedom structure subjected to free vibration and a sudden load, employed to demonstrate the physical significance of the proposed mutative-scale symbolic time-series analysis. According to the mode superposition method [28], a response vector $u\left(t\right)$ can be expressed as
(10)$$u\left(t\right)=\sum_{k=1}^{F}\Phi_{k}Y_{k}\left(t\right)\;\;\;\left(k=1,2,\ldots ,F\right),$$
where $\Phi_{k}$ is the *k*th modal shape and $Y_{k}\left(t\right)$ is a vector defined on the *k*th modal coordinate system. The latter can be written as
(11)$$Y_{k}\left(t\right)=A_{k}\sin\left(\omega_{Dk}t+\phi_{k}\right)e^{-\zeta \omega_{k}t},$$
where $A_{k}$ is the initial amplitude, ϕ is the initial phase angle, $\omega_{k}$ is the *k*th natural frequency, ζ is the damping ratio, and $\omega_{Dk}$ is the *k*th damped natural frequency, which can be expressed as
(12)$$\omega_{Dk}=\omega_{k}\sqrt{1-\zeta^{2}}.$$

Substituting Equations (Equation 11) and (Equation 12) into (Equation 10), the response $u_{i}\left(t\right)$ at the *i*th node can be obtained as follows:(13)$$\begin{array}{cc} u_{i}\left(t\right)\; & =\Phi_{1i}Y_{1}\left(t\right)+\Phi_{2i}Y_{2}\left(t\right)+\ldots +\Phi_{Fi}Y_{F}\left(t\right)\\ \; & =\Phi_{1i}A_{1}\sin\left(\omega_{D1}t+\phi_{1}\right)e^{-\zeta \omega_{1}t}+\Phi_{2i}A_{2}\sin\left(\omega_{D2}t+\phi_{2}\right)e^{-\zeta \omega_{2}t}\\ \; & +\ldots +\Phi_{Fi}A_{F}\sin\left(\omega_{DF}t+\phi_{F}\right)e^{-\zeta \omega_{F}t}. \end{array}$$

The mutative-scale interval allows for the choice of symbols over a certain period based on the damped natural period $T_{Di}$. Let the period *T* of the mutative-scale symbolic analysis method be as defined: (14)$$T=\lambda_{1}T_{D1}+\lambda_{2}T_{D2}+\ldots +\lambda_{F}T_{DF},$$
(15)$$T_{Dk}=\frac{2\pi }{\omega_{Dk}}\;\;\;\left(k=1,2,\ldots ,F\right),$$
where $T_{Dk}$ is the *k*th damped natural period and $\lambda_{k}$ is the constant associated with $T_{Dk}$. One of the sequences can be expressed as $[u_{i}\left(t_{0}\right)$,$u_{i}\left(t_{1}\right)$,…,$u_{i}\left(t_{l}\right)$,…,$u_{i}\left(t_{L-1}\right)]$ with a length *L*. Then,
(16)$$\begin{array}{cc} u_{i}\left(t_{0}\right)\; & =\Phi_{1i}A_{1}\sin\left(\omega_{D1}t+\phi_{1}\right)e^{-\zeta \omega_{1}t_{0}}+\Phi_{2i}A_{2}\sin\left(\omega_{D2}t_{0}+\phi_{2}\right)e^{-\zeta \omega_{2}t_{0}}\\ \; & +\ldots +\Phi_{Fi}A_{F}\sin\left(\omega_{DF}t_{0}+\phi_{F}\right)e^{-\zeta \omega_{F}t_{0}}, \end{array}$$
(17)$$\begin{array}{cc} u_{i}\left(t_{1}\right)\; & =u_{i}\left(t_{0}+T\right)=u_{i}\left(t_{0}+\lambda_{1}T_{D1}+\lambda_{2}T_{D2}+\ldots +\lambda_{F}T_{DF}\right)\\ \; & =\Phi_{1i}A_{1}\sin\left[\omega_{D1}\left(t_{0}+\lambda_{1}T_{D1}+\lambda_{2}T_{D2}+\ldots +\lambda_{F}T_{DF}\right)+\phi_{1}\right]e^{-\zeta \omega_{1}t_{1}}\\ \; & +\Phi_{2i}A_{2}\sin\left[\omega_{D2}\left(t_{0}+\lambda_{1}T_{D1}+\lambda_{2}T_{D2}+\ldots +\lambda_{F}T_{DF}\right)+\phi_{2}\right]e^{-\zeta \omega_{2}t_{1}}+\ldots \\ \; & +\Phi_{Fi}A_{F}\sin\left[\omega_{DF}\left(t_{0}+\lambda_{1}T_{D1}+\lambda_{2}T_{D2}+\ldots +\lambda_{F}T_{DF}\right)+\phi_{F}\right]e^{-\zeta \omega_{F}t_{1}}\\ \; & =\Phi_{1i}A_{1}\sin\left(\omega_{D1}t_{0}+2\pi \lambda_{1}+2\pi \lambda_{2}\frac{\omega_{D1}}{\omega_{D2}}+\ldots +2\pi \lambda_{F}\frac{\omega_{D1}}{\omega_{DF}}+\phi_{1}\right)e^{-\zeta \omega_{1}t_{1}}\\ \; & +\Phi_{2i}A_{2}\sin\left(\omega_{D2}t_{0}+2\pi \lambda_{1}\frac{\omega_{D2}}{\omega_{D1}}+2\pi \lambda_{2}+\ldots +2\pi \lambda_{F}\frac{\omega_{D2}}{\omega_{DF}}+\phi_{2}\right)e^{-\zeta \omega_{2}t_{1}}+\ldots \\ \; & +\Phi_{Fi}A_{F}\sin\left(\omega_{DF}t_{0}+2\pi \lambda_{1}\frac{\omega_{DF}}{\omega_{D1}}+2\pi \lambda_{2}\frac{\omega_{DF}}{\omega_{D2}}+\ldots +2\pi \lambda_{F}+\phi_{2}\right)e^{-\zeta \omega_{F}t_{1}}. \end{array}$$

The ratio $\alpha_{nk}$ of the *n*th damped natural frequency to the *k*th damped natural frequency can be defined as
(18)$$\alpha_{nk}=\frac{\omega_{Dn}}{\omega_{Dk}}.$$

The phase angle $\theta_{n}$ is
(19)$$\theta_{n}=\omega_{Dn}t_{0}+\phi_{n}.$$

Equations (Equation 17)–(Equation 19) can be used to obtain the sequence $[u_{i}\left(t_{0}\right)$, $u_{i}\left(t_{1}\right)$, …, $u_{i}\left(t_{l}\right)$, …, $u_{i}\left(t_{L-1}\right)]$, which can be expressed as
(20)$$u_{i}\left(t_{0}\right)=\Phi_{1i}A_{1}\sin\left(\theta_{1}\right)e^{-\zeta \omega_{1}t_{0}}+\Phi_{2i}A_{2}\sin\left(\theta_{2}\right)e^{-\zeta \omega_{2}t_{0}}+\ldots +\Phi_{Fi}A_{F}\sin\left(\theta_{F}\right)e^{-\zeta \omega_{F}t_{0}},$$
(21)$$\begin{array}{cc} u_{i}\left(t_{1}\right)\; & =\Phi_{1i}A_{1}e^{-2\pi \zeta \sum_{k=1}^{F}\lambda_{k}\alpha_{1k}}\sin\left(\theta_{1}+2\pi \sum_{k=1}^{F}\lambda_{k}\alpha_{1k}\right)e^{-\zeta \omega_{1}t_{0}}\\ \; & +\Phi_{2i}A_{2}e^{-2\pi \zeta \sum_{k=1}^{F}\lambda_{k}\alpha_{2k}}\sin\left(\theta_{2}+2\pi \sum_{k=1}^{F}\lambda_{k}\alpha_{2k}\right)e^{-\zeta \omega_{2}t_{0}}+\ldots \\ \; & +\Phi_{Fi}A_{F}e^{-2\pi \zeta \sum_{k=1}^{F}\lambda_{k}\alpha_{Fk}}\sin\left(\theta_{F}+2\pi \sum_{k=1}^{F}\lambda_{k}\alpha_{Fk}\right)e^{-\zeta \omega_{F}t_{0}}, \end{array}$$
(22)$$\begin{array}{cc} u_{i}\left(t_{l}\right)\; & =u_{i}\left(t_{0}+lT\right)\\ \; & =\Phi_{1i}A_{1}e^{-2\pi l\zeta \sum_{k=1}^{F}\lambda_{k}\alpha_{1k}}\sin\left(\theta_{1}+2\pi l\sum_{k=1}^{F}\lambda_{k}\alpha_{1k}\right)e^{-\zeta \omega_{1}t_{0}}\\ \; & +\Phi_{2i}A_{2}e^{-2\pi l\zeta \sum_{k=1}^{F}\lambda_{k}\alpha_{2k}}\sin\left(\theta_{2}+2\pi l\sum_{k=1}^{F}\lambda_{k}\alpha_{2k}\right)e^{-\zeta \omega_{2}t_{0}}+\ldots \\ \; & +\Phi_{Fi}A_{F}e^{-2\pi l\zeta \sum_{k=1}^{F}\lambda_{k}\alpha_{Fk}}\sin\left(\theta_{F}+2\pi l\sum_{k=1}^{F}\lambda_{k}\alpha_{Fk}\right)e^{-\zeta \omega_{F}t_{0}}. \end{array}$$

Let
(23)$$\mu_{n}=2\pi \sum_{k=1}^{F}\lambda_{k}\alpha_{nk},$$
(24)$$\begin{array}{cc} y_{n}\left(t_{l}\right)\; & =\Phi_{ni}A_{n}e^{-2\pi l\zeta \sum_{k=1}^{F}\lambda_{k}\alpha_{nk}}\sin\left(\theta_{n}+2\pi l\sum_{k=1}^{F}\lambda_{k}\alpha_{nk}\right)e^{-\zeta \omega_{n}t_{0}}\\ \; & =\Phi_{ni}A_{n}e^{-\mu_{n}l\zeta }\sin\left(\theta_{n}+\mu_{n}l\right)e^{-\zeta \omega_{n}t_{0}}, \end{array}$$

$[u_{i}\left(t_{0}\right)$,$u_{i}\left(t_{1}\right)$,…,$u_{i}\left(t_{l}\right)$,…,$u_{i}\left(t_{L-1}\right)]$ can be expressed as
(25)$$\begin{array}{cc} \; & u_{i}\left(t_{1}\right)=y_{1}\left(t_{1}\right)+y_{2}\left(t_{1}\right)+\ldots +y_{n}\left(t_{1}\right)+\ldots +y_{F}\left(t_{1}\right)\\ \; & u_{i}\left(t_{2}\right)=y_{1}\left(t_{2}\right)+y_{2}\left(t_{2}\right)+\ldots +y_{n}\left(t_{2}\right)+\ldots +y_{F}\left(t_{2}\right)\\ \; & \ldots \\ \; & u_{i}\left(t_{l}\right)=y_{1}\left(t_{l}\right)+y_{2}\left(t_{l}\right)+\ldots +y_{n}\left(t_{l}\right)+\ldots +y_{F}\left(t_{l}\right)\\ \; & \ldots , \end{array}$$

From $t_{l-1}$ to $t_{l}$, $y_{n}\left(t_{l-1}\right)$ and $y_{n}\left(t_{l}\right)$ can be expressed as
(26)$$\begin{array}{cc} \; & y_{n}\left(t_{l-1}\right)=\Phi_{ni}A_{n}e^{-\mu_{n}\left(l-1\right)\zeta }\sin\left(\theta_{n}+\mu_{n}\left(l-1\right)\right)e^{-\zeta \omega_{n}t_{0}}\\ \; & y_{n}\left(t_{l}\right)=\Phi_{ni}A_{n}e^{-\mu_{n}l\zeta }\sin\left(\theta_{n}+\mu_{n}l\right)e^{-\zeta \omega_{n}t_{0}}. \end{array}$$

The following results are obtained by comparing $y_{n}\left(t_{l-1}\right)$ with $y_{n}\left(t_{l}\right)$. Correspondingly, from $t_{l-1}$ to $t_{l}$, the amplitude is reduced by the term $e^{-\mu_{n}\zeta }$, and the phase angle is incremented by the distance $\mu_{n}$, which is associated with the constant $\lambda_{k}$ and the natural frequency of the system itself.

### 3.2. Three-Order Square Matrix

We discretize the response $u_{i}\left(t\right)$ at the *i*th node based on a certain sampling frequency into a symbolic series $S=\{s_{1},s_{2},\ldots ,s_{k},\ldots ,s_{N}\},k=1,2,\ldots ,N-\tau$, where *N* is the length of the series and N≫τ. We divide the series into *m* partitions according to the steps of the mutative-scale symbolic method. The period *T* becomes the interval τ between the chosen discrete points, that is, $\tau =f_{s}\times T$.

The number of the data from *S* falling in the *i*th partition is denoted by $a_{i}$, with i=1,2,…,m. For all the possible element pairs $\left(s_{k},s_{k+\tau }\right)$, the number whereby $s_{k}$ and $s_{k+\tau }$ relatively belong to the *i*th and *j*th partitions at the same time is denoted by $a_{ij}$, so that we obtain the mutative-scale symbolic matrix A related to τ with the size of m×m. Each entry in the matrix $A\left(\tau \right)$ is represented as $a_{ij},i,j=1,2,\ldots ,m$.

The matrix contains all the information about the changes in the inherent properties of the structure along with damage [29], and the choice of τ is of vital importance, which can be achieved through the autocorrelation coefficient, which is written as
(27)$$\begin{array}{cc} C\left(\tau \right)\; & =\frac{1}{N-\tau }\sum_{k=1}^{N-\tau }s_{k}s_{k+\tau }\\ \; & =\frac{1}{N-\tau }\sum_{i,j=1}^{m}a_{ij}\left(\tau \right)\times \left[s_{min}+\left(i-0.5\right)\times \frac{s_{max}-s_{min}}{m}\right]\\ \; & \times [s_{min}+\left(j-0.5\right)\times \frac{s_{max}-s_{min}}{m}], \end{array}$$
where $\left[s_{min}+\left(i-0.5\right)\times \frac{s_{max}-s_{min}}{m}\right]$ is the average value of the data falling in the *i*th partition.

Let $x_{ij}=\left[s_{min}+\left(i-0.5\right)\times \frac{s_{max}-s_{min}}{m}\right]\times \left[s_{min}+\left(j-0.5\right)\times \frac{s_{max}-s_{min}}{m}\right]$, representing the entries of matrix X.

As is known that $a_{ij}$ is a positive integer, Equation (Equation 27) can be written as
(28)$$\left(N-\tau \right)C\left(\tau \right)=\sum_{i,j=1}^{m}a_{ij}x_{ij}\leq \sum_{i,j=1}^{m}a_{ij}\ldots^{2}|x_{ij}|\leq \left(\sum_{i,j=1}^{m}a_{ij}\ldots^{2}\right)\left(\sum_{i,j=1}^{m}\left|x_{ij}\right|\right)={∥A∥}_{F}^{2}∥X∥,$$
where $∥X∥=\sum_{i,j=1}^{m}\left|x_{ij}\right|$ turns out to be a norm of matrix X.

As mentioned above, N≫τ, and X has nothing to do with τ, so the upper bound of $C\left(\tau \right)$ is closely related to the Frobenius norm FN of $A\left(\tau \right)$, $FN={∥A∥}_{F}$, which can be an indicator of the varying system [30].

In conclusion, it can be proven that when damage occurs to a structure, the inherent properties of the structure change, and the ratio of the natural frequency of the structure changes accordingly. As the degree of damage increases, there must be more than one set of constant $\lambda_{k}$ that makes the damage indicator associated with τ increase continuously. In other words, the mutative-scale symbolic matrix method with an appropriate interval τ can reflect the damaged state of the structure. By the way, the choice of τ is not unique.

## 4. Abaqus Simulation

### 4.1. Description of Abaqus Model

A simply supported beam [31] with dimensions of length l= 5 m, height h= 0.5 m, and width w= 0.01 m is simulated by using Abaqus Explicit finite element code to verify the proposed damage identification method. A total of 15 sensors are in the vertical direction at the top of the beam, as shown in Figure 2. The damage is represented as a single vertical crack at the bottom of the beam corresponding to the position of sensor 8, with different crack lengths of 10, 20, 30, 50, 100, 150, and 200 mm, which represent eight damage cases, including the undamaged state, in Table 1. The contact of the crack surfaces is assumed to be frictionless in the tangential direction. A random load of lateral movement is applied to the top surface of the beam, which is different in each case. Noise is added to the acceleration records obtained from the finite element analyses. The signal-to-noise ratio (SNR) is 30 dB, which is a typical value in vibration measurement systems [32].

### 4.2. Damage Identification Procedure of Abaqus Model

The measurement period is two seconds including 4001 samples, and 10 samples of each damage case are chosen randomly to verify the effectiveness of the damage identification method. In the process, the damage is affirmed to occur between the virtual input and the output positions. The following are further assumed:(1)The accelerated response of the 3rd sensor is used as the virtual input, the accelerated response of the 13th sensor is used as the output to calculate the TF, and the VFRF H(t) is obtained through the inverse Fourier transform, as shown in Figure 3a. As can be observed, with the increase in damage, the attenuation of the VFRF gradually becomes stronger, which means that the damage has an impact on the characteristics of the beam. It is worth noting that the selection of the accelerated response points is not unique but will influence the movement of the autocorrelation coefficient curves.(2)Then, the autocorrelation coefficient is plotted, and the concentrated extreme points are found as shown in Figure 3b with the appropriate τ corresponding to them. The extreme points of the Pearson autocorrelation coefficients correspond to the intervals of 5, 10, 15, and 20, and the 3×3 matrices $A\left(\tau \right)$ are established based on the intervals, along with the Frobenius norm as damage indicators.

### 4.3. Simulation Results

The damage indicators at the intervals of 5, 10, 15, and 20 are shown in Figure 4. In the figure, the abscissas represent eight damage cases, and the ordinates represent the damage indicator. There are five gray curves in each subfigure representing five independent replicates and one red line representing the mean value. The value of the damage indicator does not change much under the first five damage cases but suddenly increases after the fifth case, which is consistent with the crack length of each damage case (10, 20, 30, 50, 100, 150, and 200 mm). When the crack is small, sometimes, the indicator cannot accurately capture the damage deterioration due to interference. The trend is more obvious after the fifth case, because when the crack is larger than 50, the length of the crack in adjacent cases increases significantly, and the value of the indicator follows closely as the crack grows rapidly. On the whole, the damage identification method with the indicator can reflect the damaged condition of the structure with good robustness.

## 5. Experiment in Laboratory

### 5.1. Description of Experimental Structure

The wooden truss bridge experiment, conducted by Kullaa et al. [33], is applied as further proof of damage identification, and the experimental setup is shown in Figure 5. In the laboratory, under a changing environment (temperature, humidity, etc.), a long-term dynamic test of the structure is monitored by 15 acceleration sensors, using white noise as excitation, to measure the acceleration response at 15 positions of the structure, with the sampling frequency set to 256 Hz. The positions of the sensors are marked as shown in the figure. To simulate the damage of the 36 kg weight structure, a series of mass blocks, weighing from 23.5 g to 193.66 g, are attached to the beam between sensors 1 and 2, to form six damage cases along with damage aggravation, as shown in Table 2.

### 5.2. Damage Identification Procedure of Experiment

The acceleration response of sensor 1 is selected as the virtual excitation, and the acceleration response of sensor 15 is selected as the output for damage identification. There are about 20 groups of independent data of each acceleration sensor in each case, 5 of which are selected randomly as five independent repeated trials, to verify the effectiveness and robustness of this method. The VFRF of the structure is shown in Figure 6a. The attenuation is not obvious in the figure, indicating that this type of damage cannot significantly alter the structural characteristics. The autocorrelation coefficient is plotted in Figure 6b; there are many extreme points in the autocorrelation coefficient curve. We choose an area where the extreme points are relatively concentrated, corresponding to the abscissa values within 50–80, and select the non-adjacent extreme points. It is found through trial calculation that the low test sampling frequency may lead to coordinate deviations of τ, namely, the abscissas of the extreme points may occur between two adjacent integers. Under this premise, it is proven that the damage condition can be better reflected at the intervals of 58, 70, and 78, which are taken as references to verify the damage identification of the structure. It is worth noting that other extreme points may still have the same effect. The damage indicator is still the Frobenius norm of the matrix $A\left(\tau \right)$.

### 5.3. Experimental Results

The structural damage identification results are shown in Figure 7 when the considered intervals are 58, 70, and 78. In Figure 7a, the horizontal axis represents the six damage cases, while the vertical axis represents the damage indicator FN. There are still five trials and their mean value in each figure. The damage indicator curves generally increase with the increase in damage (the added mass values between sensors 1 and 2 are 0, 23.5 g, 47 g, 70.5 g, 123.16 g, and 193.66 g). Therefore, the proposed method with damage indicators can be used to perform damage identification.

In Figure 7b, the horizontal axis represents five independent repeated tests, and the vertical axis still represents the damage indicator FN. There are six curves in the figure, representing the damage indicator in different damage cases. Among them, the bottom curve is the value of the original state, the top curve is the value of the highest degree of damage, and the other curves are listed successively with the increase in damage. From the figures, the damage indicator curves have tiny fluctuations under five different excitations, and some small twists may be caused by interference in the laboratory but are still acceptable. So, the damage identification method is proven to be effective and has a certain degree of robustness. Therefore, even if this type of damage cannot significantly alter the structural characteristics, the proposed method can still accurately identify the occurrence of damage.

## 6. Structure in Service

### 6.1. Description of Practical Structure

Tianjin Yonghe Bridge [34], located in the eastern suburbs of Tianjin, was built in 1983 and opened to traffic in December 1987. It is one of the earliest cable-stayed bridges in China, as shown in Figure 8a. The total length of the bridge is 512.4 m, the main span is 260 m, and the two side spans are 25.15 m and 99.85 m. The full width of the bridge deck is 13.6 m, including 9 m for lanes and 1 m for sidewalks on both sides. The height of the two towers is 60.5 m. The main girder of the bridge is composed of 74 precast concrete beams, and two 41.6 m prestressed beams straddle the bridge and were poured by using temporary scaffolding. The whole bridge is equipped with 88 pairs of stay cables, each of which is wound by steel wire with a diameter of 5 mm. The number of steel wires in the cable is at least 69 and at most 199.

In 2007, after 19 years of service of Tianjin Yonghe Bridge, several cracks were found on the lower surface of a main beam in the middle span of the bridge, among which the maximum width was 2 cm, as shown in Figure 8b. At the same time, serious corrosion occurred near the anchorage position of the stay cable. It is speculated that the cracks have been caused by the bridge’s long-term overweight operation because the weight and traffic flow of vehicles passing over the bridge are much higher than expected when the bridge was designed. To ensure the safety of the entire bridge, local authorities conducted overall maintenance and overhaul between 2005 and 2007, including repouring the mid-span main beams and replacing all the stay cables.

In the process of overhaul and maintenance of Yonghe Bridge [35], the Structural Monitoring and Control Research Center of Harbin Institute of Technology designed and laid a set of complex structural health monitoring systems for the bridge. More than 150 sensors with various functions are deployed throughout the bridge, including the stay cables, pylons, and main beams. Fourteen uniaxial acceleration sensors are installed on the bridge panel, and one biaxial acceleration sensor is installed on the top of the south pylon to monitor the horizontal vibration of the pylon. In addition, an anemometer and a temperature sensor are installed on the top of the south tower to monitor the three-dimensional wind speed of the bridge and the ambient temperature changes, respectively. A weightier-in-motion (WM) system was also installed on the bridge panels to monitor vehicles in all bridge lanes in real time. In addition, several fiber Bragg grating sensors are embedded in the mid-span main beam to monitor the strain and temperature changes at the key points of the bridge. The arrangement of sensors in the whole bridge is shown in Figure 9. All numbers in the figure are in centimeters.

### 6.2. Damage Identification Procedure of Bridge

The acceleration data of Yonghe Bridge are not recorded every day by mainly on 3 February, 19 March, 30 March, 9 April, 5 May, 18 May, 31 May, 7 June, 16 June, and 31 July. Data are collected for 24 h each day, at a sampling frequency of 100 Hz. According to research by Huang et al. [36], damage is most likely to occur on 5 May and 18 May, so we choose the dates of 9 April, 5 May, and 18 May for verification. To ensure data reliability, the 10 sets of acceleration responses of sensor 2 in the first two hours of the three days are selected as virtual excitation, while the acceleration responses of sensor 12 are selected as the outputs for damage identification. The 10 sets of data are considered 10 independent repeated trials to verify the effectiveness and robustness of this method, and the autocorrelation coefficient is shown in Figure 10a. Then, for detection, we further use the acceleration responses of sensor 3 of the last two hours of the three days as the virtual excitation and the acceleration responses of sensor 13 as the output, as shown in Figure 10b. There are many extreme points in the autocorrelation coefficient curves, and the most obvious points (21, 38 and 56, 86) are chosen for validation.

### 6.3. Experimental Results

Figure 11a,b show the results of sensor 2 and sensor 12, which represent the development of damage indicators from 9 April to 5 May and to 18 May. The curves of damage indicators for the three days are a little disordered due to interference, so more attention is paid to the mean value of the ten trials of the indicators, which can reflect the damage development of the structure. The starting and end points of the mean values are adjusted to the appropriate position in the plots to compare the movements of the values, and it is observed that the curve continues to climb in two periods, which means that the damage does become serious over time. However, the change in value is minimal, namely, damage changes little on the whole, which is in line with the actual situation. As shown in Figure 11c,d, for sensor 3 and sensor 13, the measurements were taken in the evening. The trend of the damage indicators is basically consistent with that for sensors 2/12, but it is found that the increase in damage indicators on 18 May to 5 May is much higher than that in Figure 11a,b, which makes it certain that the degree of damage changed considerably on 18 May.

Compared with the method by Huang et al. [36], who referenced the results of three algorithms to screen the most likely time of damage occurrence, the proposed method can more intuitively reflect the day when damage is likely to occur. In addition, when using the stopwatch timer in Matlab R2021a to calculate the time consumed, the proposed method can save more than half of computation time because of the application of intermittent points and matrices. It can be foreseen that more accurate damage details can be obtained if abundant data are used for further research.

## 7. Conclusions and Discussion

### 7.1. Conclusions

A new damage identification method based on STSA and TF is proposed to solve the difficulty of structural damage identification under ambient excitation. Compared with the conventional STSA method, which has limitations in data processing, the new method uses the autocorrelation coefficient and the maximum entropy principle to select the appropriate interval to extract the most valuable data and combines the data into a three-order square matrix following a new sampling pattern, with the norm of the matrix as the damage indicator for damage identification. The theoretical derivation in this study proves that this method reduces the dimension of the original data, improves the accuracy of damage identification, and reduces the amount of calculation. A simulation model in Abaqus of a simply supported beam, an experiment in the laboratory on a wooden truss bridge, and the actual engineering scenario of a cable-stayed bridge are together employed to validate the accuracy and reliability of the proposed method. In addition, the proposed method exhibits superior robustness to different damage types.

In this process, the TF plays a role in eliminating the impact of varying excitation. Although the definition of TF is only aimed at structures under single-point excitation, the combination of the new sampling pattern in this study makes natural incentives equally applicable according to the results of the examples of Abaqus, laboratory, and practical engineering under different types of excitation, which validates that the proposed method is always feasible and has a certain degree of robustness to environmental changes. Compared with other damage detection methods, this method still embodies its advantages of intuitiveness and ease of calculation in structural damage identification under ambient excitation.

### 7.2. Limitations and Prospects

The proposed damage detection method has good prospects in more complex structures with large datasets for its significant improvement in calculation efficiency and robustness to different types of damage. In this paper, the proposed method is applied only to one-dimensional structures that can be simplified into chain models to ensure that damage occurs between sensors. Therefore, for large spatial structures, the optimized arrangement of sensors is of vital importance.

The arrangement of sensors is also crucial in damage localization. At present, damage localization in simple chain-like structures can be performed, but for complex structures, it is still under study and will be shared in subsequent studies.

## Figures and Tables

**Figure 1 sensors-24-05941-f001:**
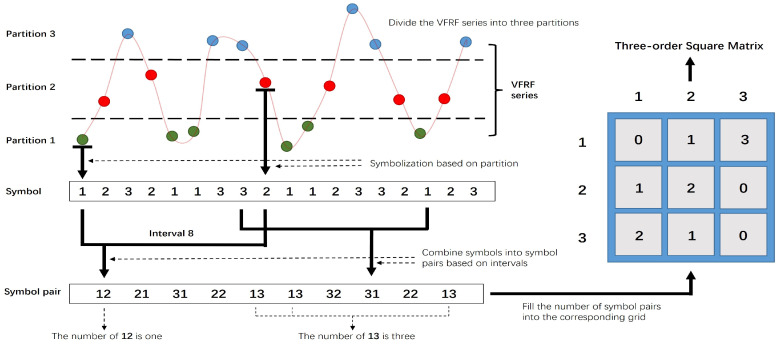
The new sampling pattern to form the three-order square matrix.

**Figure 2 sensors-24-05941-f002:**
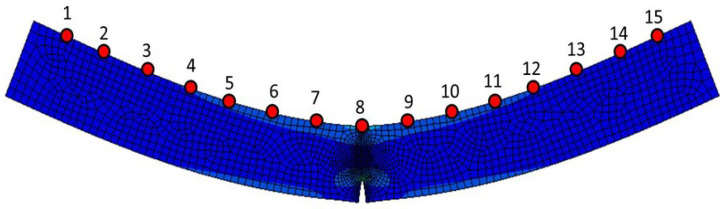
The grid diagram of the structure in Abaqus [31].

**Figure 3 sensors-24-05941-f003:**
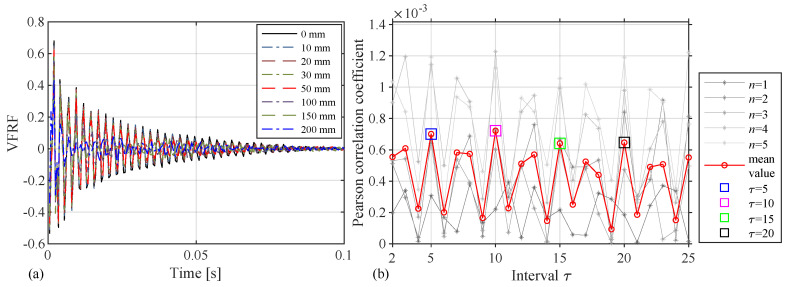
The simulation of the Abaqus model: (**a**) VFRF and (**b**) autocorrelation coefficient.

**Figure 4 sensors-24-05941-f004:**
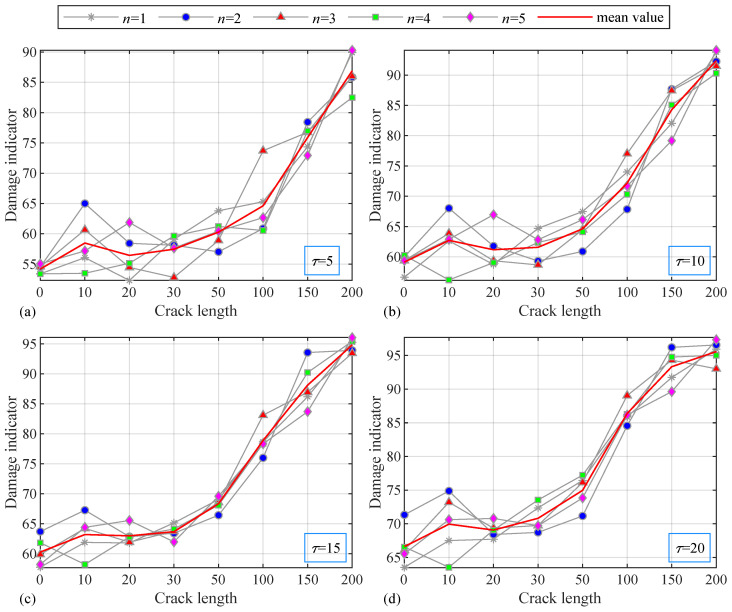
Damage indicators of eight cases with (**a**) τ=5; (**b**) τ=10; (**c**) τ=15; (**d**) τ=20.

**Figure 5 sensors-24-05941-f005:**
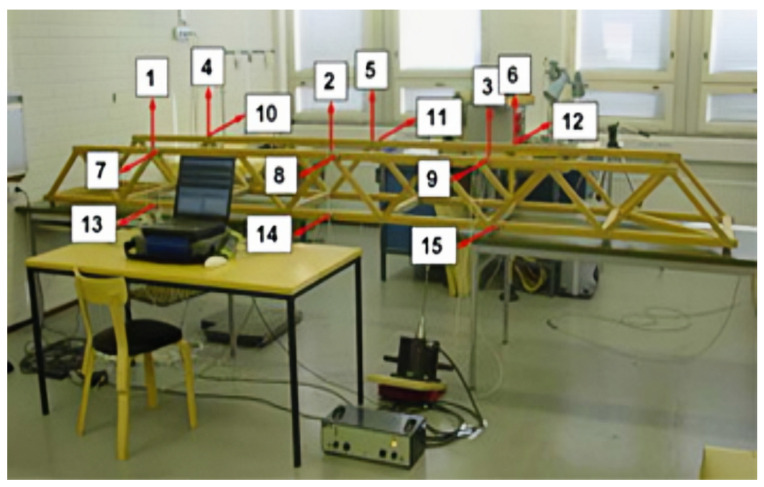
The schematic of the wooden truss bridge [33].

**Figure 6 sensors-24-05941-f006:**
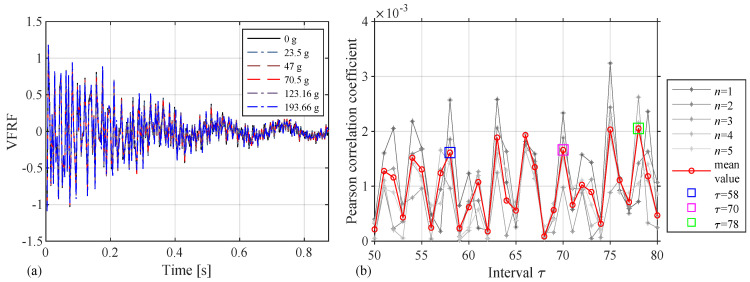
The experiment of the wooden truss bridge: (**a**) VFRF and (**b**) autocorrelation coefficient.

**Figure 7 sensors-24-05941-f007:**
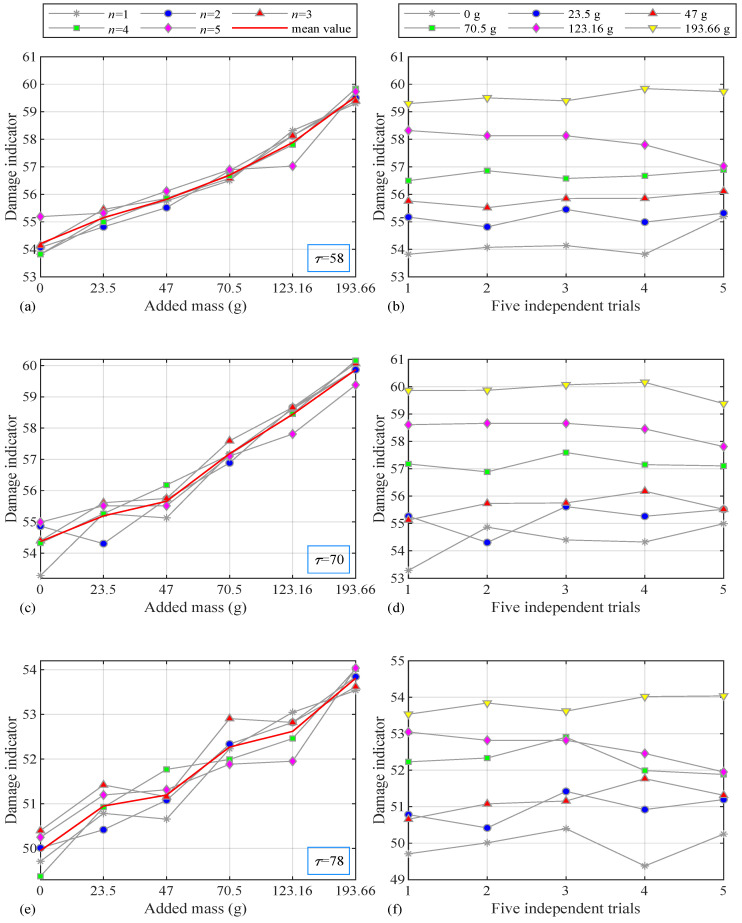
Damage indicators of six cases and five independent trials with (**a**,**b**) τ=58; (**c**,**d**) τ=70; (**e**,**f**) τ=78.

**Figure 8 sensors-24-05941-f008:**
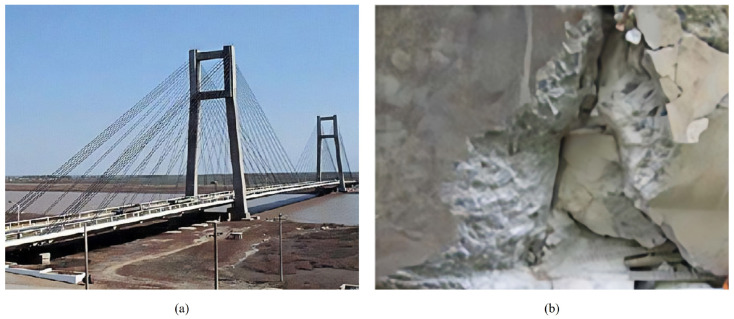
The photos of the structure in service: (**a**) overview of Tianjin Yonghe Bridge and (**b**) cracks in the middle span of the bridge [34].

**Figure 9 sensors-24-05941-f009:**
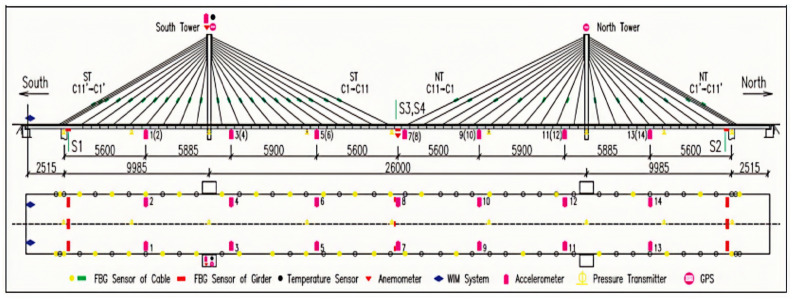
Sensor settings of Tianjin Yonghe Bridge SHM system [34].

**Figure 10 sensors-24-05941-f010:**
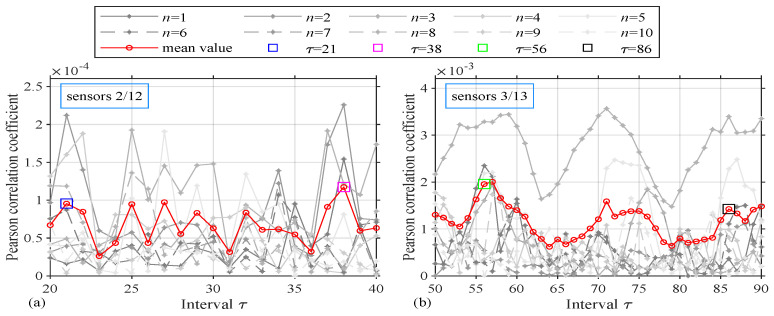
Tianjin Yonghe Bridge: (**a**) Autocorrelation coefficients for sensors 2/12 and (**b**) autocorrelation coefficients for sensors 3/13.

**Figure 11 sensors-24-05941-f011:**
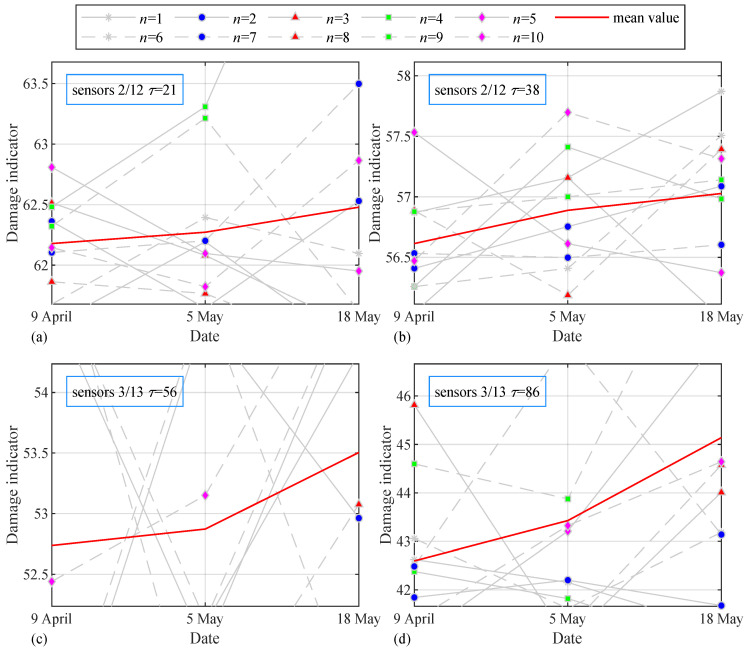
Damage indicators for sensors 2/12 with (**a**) τ=21; (**b**) τ=38; damage indicators for sensors 3/13 with (**c**) τ=56; (**d**) τ=86.

**Table 1 sensors-24-05941-t001:** Three damage cases of the Abaqus model.

Damage Condition	Crack Length (mm)
Case 1	0
Case 2	10
Case 3	20
Case 4	30
Case 5	50
Case 6	100
Case 7	150
Case 8	200

**Table 2 sensors-24-05941-t002:** Six damage cases of the experimental structure.

Damage Condition	Added Mass (g)
Case 1	0
Case 2	23.5
Case 3	47
Case 4	70.5
Case 5	123.16
Case 6	193.66

## Data Availability

Data are contained within the article.

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
