# Peer review of "Structural Damage Detection under Ambient Excitation Using Symbolic Three-Order Square Matrix Formed by Specific-Interval-Sampled Time-Domain Signals"

_sensors, 2024, doi:10.3390/s24185941_

Round 1

Reviewer 1 Report

Comments and Suggestions for Authors

This manuscript reports on a novel methodology to extract damage information from acceleration data, by a combination of a series of postprocessing techniques to the original raw data signal, including analyzing the data in specific intervals using Pearson autocorrelation methods. An innovative process of converting extracted points into matrix entries is then applied such that this matrix norm is the measurement of damage. The manuscript is well written and in acceptable quality. However, I have some minor concerns that I would like the authors to address before I recommend this manuscript for publication.

1. The literature review needs further explanation of relevant findings in the fields of interest. The authors just provide lumped references to the general topics covered  without specifying details. The authors should clearly establish the state-of-the-art in the field through explaining findings of similar case studies by other researchers in a specific way (e.g., "Researcher X proposed the use of YYY method and demonstrated its performance in ZZZ experiment whose conclusions were WWWW".

2. Explanation of Fig. 2 is not clear. The authors state that the "attenuation of FRF gradually becomes stronger with the increase of damage", and that this is intuitively observed in the figure. This is not intuitively observed in the figure. It is not clear what the legend values mean. (e.g., 0, 10, 20, ...). It was assumed that these are the damage levels. If this is the case, the figure needs to have higher quality and resolution to offer strong evidence for this statement. The quality and resolution comment also applies for Fig. 6.

3. The process of obtaining a time-domain response for 0.1 seconds is also not clearly explained. Even though an expert reader can figure out this postprocessing artifice, I suggest the authors to provide a clean flow diagram of all the procedures/techniques used during the damage ID process. I can think of a similar diagram recently published in https://doi.org/10.1115/1.4065777 (Fig. 10), just for reference of a similar diagram that can significantly help readers understand the methods used.

4. Fig. 5 is exactly extracted from a publication in other journal and not cited in the caption. This may imply a copyright problem. Consider using unpublished material. Also, the numbers in the figure are not explained anywhere in the document. 

5. There is no mention of the advantages of using this proposed methodology against the use of traditional methods, widely in use by the community. The authors should carefully address the question Why would researchers use this approach over existing ones? Perhaps the processing time or computation cost play a role in answering this question. Please consider selling your story in a stronger way.

I have also made some observations in the attached revised PDF that I would like the authors to address, particularly regarding the quality of the figures and other minor edits.

Comments on the Quality of English Language

There are minor typo and clarification edit suggestions in the attached commented PDF

Reviewer 2 Report

Comments and Suggestions for Authors This paper presents a method for structural damage detection under ambient excitation, utilizing a symbolic three-order square matrix formed by specific-interval sampled time domain signals. The study demonstrates the effectiveness of this approach through theoretical analysis and experimental validation. My comments are: 1. Some of the charts are not sufficiently clear, such as Figure 1, which lacks detailed explanatory text. Please provide more comprehensive explanatory text to clarify each step in the chart. 2. In lines 280 and 281, How is the crack length selected, and what are the bases for the selection? 3. Please discuss the limitations of the method proposed in this paper, including but not limited to potential sources of error and situations where the method may not achieve optimal performance. 4. Several relative studies should be mentioned: Optimal static strain sensor placement for truss bridges. Damage identification of large-scale space truss structures based on stiffness separation method. 5. Could you provide an overview of potential directions for future research in the conclusion section, such as applying this method to larger datasets or more complex structures? 6. In line 258, the Pearson autocorrelation coefficient is mentioned. For those not very familiar with the Pearson autocorrelation coefficient, understanding it can be somewhat challenging. Could you please provide a detailed explanation of the Pearson autocorrelation coefficient?

7. In lines 325 and 326, the quality block is placed between sensors 1 and 2 on the beam. Is it positioned at the midpoint or the quarter points? Would putting it at different locations affect the results?

Round 2

Reviewer 1 Report

Comments and Suggestions for Authors

The authors have addressed all the concerns I had with the original manuscript.

Reviewer 2 Report

Comments and Suggestions for Authors

The manuscript has been revised and the quality has been improved.